

# The prevalence and associated factors of sleep deprivation among healthy college students in China: a cross-sectional survey

Congcong Guo[1,*], Songzhe Piao[2,*], Chenyu Wang[1], Lili Yu[3], Kejun Wang[1], Qian Qu[1], Cuiting Zhang[1] and Xiaofei Yu[1]

[1] School of Integrated Traditional Chinese and Western Medicine, Binzhou Medical University, Yantai, Shandong, China
[2] Department of Urology, Taizhou Hospital of Zhejiang Province Affiliated with Wenzhou Medical University, Linhai, Zhejiang, China
[3] Faculty of Chinese Medicine, Macau University of Science and Technology, Macao, China
[*] These authors contributed equally to this work.

Corresponding author
Xiaofei Yu, yuxiaofei0206@qq.com

## ABSTRACT

**Background**. The prevalence of sleep deprivation among college students is increasing and has a few associated factors.

**Methods**. The present study analyzed 2,142 college students from 28 provinces in China. The Chinese version of the Pittsburgh Sleep Quality Index (PSQI) was used to assess sleep duration. Binary logistic regression was conducted to explore the sleep deprivation related factors. Age and gender were controlled as covariates.

**Results**. Among the 2,142 college students (27.7% male, 72.3% female), 1,620 (75.6%) reported the average sleep duration was below 7 h per day for one month, 49.3% (1,055/2,142) slept 6∼7 h (contains 6 h), 21.0% (449/2,142) slept 5∼6 h (contains 5 h), and 5.4% (116/2,142) slept <5 h. Age increased the risk of sleep deprivation, the adjusted odds ratio = 1.05 (95% CI [1.01∼1.10]). The adjusted odds ratio (A-OR) for sleep deprivation was higher for students of more than 60 min nap duration per day (A-OR = 2.35, 95% CI [1.45∼3.80]), and age growth (A-OR = 1.05, 95% CI [1.01∼1.10]). In contrast, A-ORs were lower among sleeping inconsistency between work and rest days (A-OR = 0.61, 95% CI [0.49∼0.75]), accustomed to staying up late (A-OR = 0.45, 95% CI [0.36∼0.57]), staying up late to work or study (A-OR = 0.62, 95% CI [0.49∼0.78]), stress (A-OR = 0.75, 95% CI [0.58∼0.98]), and repeated thoughts in bed had (A-OR = 0.79, 95% CI [0.62∼0.99]).

**Conclusions**. Sleep deprivation is extremely common among healthy college students in China. It is necessary to perform methods maintaining enough sleep due to the current high incidence of sleep deprivation. Controlling the nap duration and getting enough sleep on rest days to replace missing hours of sleep on workdays might improve college students' sleep.

## INTRODUCTION

In recent years, sleep deprivation is prevalent with the increasing social competition and the convenience of entertainment and socializing. The American Academy of Sleep Medicine and the Sleep Research Society recommend that adults aged 18–60 years should have at least 7 h of sleep per night (*Watson et al., 2015*). In the United States, more than 90 million people reported that they slept less than 6 h per day (*Yong, Li & Calvert, 2017*). In 2004, 28.6% of Americans slept less than 6 h, while it grew to 32.9% in 2017 (*Sheehan et al., 2019*). Sleep deprivation occurs in any age range, and it is a growing problem in the youth population, including college students (*Kansagra, 2020*). The prevalence of sleep deprivation among college students is about 70% in the United States (N5 = 1,125, N6 = 191, N7 = 237) (*Lund et al., 2010*; *Buboltz Jr, Brown & Soper, 2001*; *Tsai & Li, 2004*), about 49.6% in Saudi Arabia (N8 = 805) (*Althakafi et al., 2019*), and about 52.7% in Lebanon (N9 = 735) (*Assaad et al., 2014*). A good night's sleep plays a vital role in maintaining body balance and promoting one's physical and mental health. Previous studies showed that lack of adequate sleep is associated with a higher risk of disease and even death (*Pan et al., 2014*; *Liu et al., 2018*). The disorders include but are not limited to, hypertension (*Matenchuk, Mandhane & Kozyrskyj, 2020*), obesity (*Larcher et al., 2015*), obstructive sleep apnea, vascular disease (*Grandner et al., 2014*), stroke, myocardial infarction, depression, anxiety, psychosis (*Sondrup et al., 2022*), attention, memory and cognitive impairment (*Kim et al., 2022*; *Newbury et al., 2021*; *Wei, Duan & Guo, 2022*). Sleep deprivation not only affects the quality of sleep but also the work status and life status (*Pires et al., 2016*). As a risk factor for a variety of disorders, it is important to prevent the incidence of sleep deprivation.

Sleep deprivation is associated with multiple factors. Previous studies have reported the factors are age (*Wei, Duan & Guo, 2022*; *Pires et al., 2016*), gender (*Zhang et al., 2008*), race/ethnicity (*Zhang et al., 2008*; *Zhang et al., 2006*; *Gangwisch et al., 2006*), dietary habits (*Zheng et al., 2016*; *Buysse et al., 1989*; *Becker et al., 2018*), lifestyle (smoking, drinking, internet/smartphone use, physical activity, and sedentary behavior) (*Zhang et al., 2008*; *Zhang et al., 2006*; *Gangwisch et al., 2006*; *Zheng et al., 2016*; *Buysse et al., 1989*), economic status (*Wei, Duan & Guo, 2022*; *Zhang et al., 2008*; *Mohammadbeigi et al., 2016*), serious illness (*Yan et al., 2021*), and sleep environment (*Mohammadbeigi et al., 2016*).

However, there are few reports on the prevalence and influencing factors of sleep deprivation among college students in China. This study aims to estimate the prevalence of sleep deprivation among healthy college students in China and to identify associated factors. Such findings may be used to highlight the issue of sleep deprivation among college students in the country and to guide future educational/behavioral/policy changes that can improve sleep sufficiency in this important subgroup of the population.

## METHODS

### Study design and the participants

The survey was conducted from July 31, 2021, to March 16, 2022. The cross-sectional survey design and snowball sampling method were conducted. The data was collected by questionnaire star, an online questionnaire survey platform anonymously. The

questionnaire was transmitted through WeChat or Tencent QQ, which were close to 100% of the applications used by Chinese college students. The participants were healthy college students from 28 provinces of China, including Beijing, Shandong, Jiangsu, Xinjiang, Ningxia, *etc.* The study excluded students who were on night shifts to avoid sleep disruptions due to night work or had sleep deprivation due to sleep-related disorders. Validation questions were set to ensure the validity of the questionnaire responses. A total of 2289 questionnaires were returned, including 147 invalid questionnaires with incorrect answers to the validation questions and more than 1/3 vacancies. In total, 2142 valid questionnaires were analysed in this study.

## Study tool

A self-assessment questionnaire including socio-demographic and sleep information was used to collect data. Socio-demographic information included age, gender, body mass index (BMI), marital status, nationality, education level, and grade. The Chinese version of the Pittsburgh Sleep Quality Index (PSQI) was used to assess sleep quality. There is good reliability and validity in the Chinese version of PSQI (*Zheng et al., 2016*; *Lopresti et al., 2020*). It can check for sleep quality in patients and healthy people (*Buysse et al., 1989*; *Yan et al., 2021*; *Akman et al., 2015*; *Yilmaz, 2020*). The PSQI, a 19-item self-reported questionnaire, measures seven areas: subjective sleep quality, sleep latency, sleep duration, habitual sleep efficiency, sleep disturbances, use of sleep medication, daytime dysfunction over the past month (*Buysse et al., 1989*). The Likert score for each category in the PSQI is 0–3. The total score is the sum of the scores of 7 aspects, ranging from 21 to 100, with higher scores indicating much worse sleep quality.

The questionnaire also contained some questions that may be related to sleep duration, including smoking, drinking, nap duration, sleep duration on workdays and rest days (same, workdays >rest days, rest days day >workdays), taking time away from sleep for recreation (yes or no), refreshing drinks (such as coffee, tea and others) interfering with sleep (yes or no), accustomed to staying up late (yes or no), staying up late to work or study (yes or no), the dormitory environment affects sleep (yes or no), relationship issues (yes or no), stressed out state (yes or no), repeated thoughts in bed (yes or no), mood problems (yes or no), desire to coordinate multiple life roles (yes or no).

## Sleep deprivation defining

Currently, there is no specific definition of sleep deprivation. According to the National Sleep Foundation (NSF), young adults aged 18-25 and adults aged 26-64 are not recommended to sleep <7 h per day (*Hirshkowitz et al., 2015*). In the present study, sleeping less than 7 h was considered as sleep deprivation.

## Health status determination

Health status was determined by the following two questions in the questionnaire. Question 1: ''Are you suffering from or have been diagnosed mental or physical illness?'' Question 2: ''Are you in a good physical and mental state currently?'' Participants were considered to be healthy if the answer to the first question is ''No'' and the answer to the second question is ''Yes''.

## Ethics

The study was conducted according to the guidelines of the Declaration of Helsinki and approved by the Ethics Committee of Binzhou Medical University (2021-262). Informed consent was given before the information collected. Data were collected anonymously.

## Sample size determination

Sample size of participants was estimated by a standardized formula: $N = Z_\alpha p(1-p)/\delta^2$. As recommended, 0.05 was set to be the significance level, $Z_\alpha$ (statistic of significance test) was 1.96, and 0.1 was the estimated acceptable margin of effort for proportion. According to a study about 18,211 undergraduate university students from 24 low—and middle-income and two high-income countries, the overall prevalence of short sleep duration was around 39% (*Peltzer & Pengpid, 2017*). The P (Prevalence) was set as 0.39 in our study. The minimum sample size for current study was 592.

## Statistical analysis

The Kolmogorov–Smirnov one-sample test was performed to examine the normality of the data set. Since the data was non-normal distribution, the continuous variables were described for the median and interquartile range, and the categorical variables were described for percentages. The participants were divided into sleep deprivation and non-sleep deprivation groups, according to sleep duration less or more than 7 h, and differences in socio-demographic and sleep-related factors were compared between the two groups by using the Mann–Whitney U test for continuous variables and the chi-square test for categorical variables. Analysis of covariance (ANCOVA) was conducted to control the effects of age and gender on sleep duration. Binary logistic regression was conducted to explore the sleep deprivation related factors. Univariate logistic regression was performed firstly. If the *p*-value was less than 0.10, the factor was included in a multivariate logistic regression model. The backward method was used to select the factors in the final logistic regression model, and an adjusted odds ratio was computed. The goodness of fit was assessed by the Hosmer-Lemeshow test, and 95% confidence interval (CI) was reported. SPSS 22.0 was conducted for all statistical analyses. *P*-values were double-tailed with a significance level of 0.05.

## RESULTS

### Sample characteristics

A total of 2,142 participants were obtained for analysis. A large proportion (55.4%, 1,187) of participants was from Shandong, but the survey was completed by respondents in 27 out of the 34 China provinces. Of the 2,142 individuals, 1,549 (72.3%) were females, and 593 (27.7%) were males. The mean (SD) age was 18.9 ± 2.3 years (range = 16–35 years). Junior college students made up the majority of the sample (59.4%), followed by undergraduates (34.9%). A considerable percentage of college students were in their freshman year (42.7%) or sophomore years (32.8%). Most of these students are ethnic Han (95.6%), and a few are of other ethnicities (4.4%). Almost all students were single (99.0%) and only a very few students were married (1.0%). Furthermore, few proportions of students reported

smoking (2.0%) and drinking behavior (11.4%). Table 1 shows the socio-demographic, smoking, and drinking characteristics of participants.

## Prevalence of sleep deprivation

Daily sleep duration of college students was 6.4 h (standard deviation (SD) = 1.0, ranging from 3 to 12 h). 75.6% (1620) college students reported sleep deprivation (sleep duration <7 h). Among 1620 sleep deprivation respondents, 1,055 (49.3%) reported a sleep duration 6∼7 h (contains 6 h), 449 (21.0%) reported 5∼6 h (contains 5 h), 116 (5.4%) reported <5 h. Furthermore, 24.4% (522) college students reported healthy sleep time: 7∼8 h (contains 7 h) have 460 (21.5%), 8∼9 h (contains 8 h) have 46 (2.1%) and ≥9 h have 16 (0.7%).

Furthermore, nap duration was common, with 953 (44.5%) students taking a nap 16∼30 min, 514 (24.0%) 31∼60 min, and 95 (4.4%) >60 min. A total of 1,242 (58.0%) students slept longer on rest days, 215 (10.0%) longer on workdays, and 685 (32.0%) were consistent.

## Factors associated with sleep deprivation

According to whether they slept <7 h, the subjects were divided into two groups: sleep deprivation and non-sleep deprivation. Two groups compared socio-demographics and sleeping characteristics, found in the two groups (sleep-deprived students and non-sleep-deprived students) had significant differences in age ($p < 0.001$), nap duration ($p < 0.001$), and sleeping duration on workdays and rest days ($p < 0.05$). Since age and gender were significantly associated with sleep duration, they were controlled as covariates and analyzed. The result showed that nap duration was also different between the sleep deprivation group and non-sleep deprivation group ($F = 4.791$, $p < 0.01$). However, there was no significant difference in BMI, education level, grade, marital status, nationality, smoking and drinking between sleep deprivation and non-sleep deprivation groups (all $p > 0.05$) with/without adjusting for the effects of the age. The details are shown in Table 2.

Then binary logistic regressions were used to examine socio-demographic and sleep-associated factors. In adjusted models, sleep deprivation was associated with age, nap duration, sleeping duration on work and rest days, accustomed to staying up late, staying up late to work or study, stressed out state, repeating thoughts in bed.

Age increased the risk of sleep deprivation (A-OR = 1.05, 95% CI [1.01∼1.10]). Compared to less than 15 min, and more than 60 min nap duration was more likely to be sleep deprivation (A-OR == 2.35, 95% CI [1.45∼3.80]). Compared with sleeping consistency, sleeping inconsistency between work and rest days had 40% (A-OR = 0.61, 95% CI [0.49∼0.75]) lower odds of sleep deprivation. Accustomed to staying up late and staying up late to work or study had 54% (A-OR = 0.45, 95% CI [0.36∼0.57]) and 37% (A-OR = 0.62, 95% CI [0.49∼0.78]) significantly lower risk of sleep deprivation compared with sleeping normally separately.

High-stress 24% (A-OR = 0.75, 95% CI [0.58∼0.98]) reduced odds of sleep deprivation. Repeated thoughts in bed had 21% (A-OR = 0.79, 95% CI [0.62∼0.99]) lower risk of sleep deprivation.

**Table 1  Sample characteristics.**

| Socio-demographic | Categories | Participants | Smoking, drinking and sleep condition | Categories | Participants |
|---|---|---|---|---|---|
| Age (Median ± SD) | – | 18.9 ± 2.3 | Smoking (n, %) | No | 2,100 (98.0%) |
| BMI (Median ± SD) | – | 21.9 ± 4.0 | | Yes | 42 (2.0%) |
| Gender (n, %) | Male | 593 (27.7%) | If smoking, the number of cigarettes smoked per day (Median ± SD) | – | 6.3 ± 7.2 |
| | Female | 1,549 (72.3%) | Drinking (n, %) | No | 1,898 (88.6%) |
| Education level (n, %) | Junior college | 1,272 (59.4%) | | Yes | 244 (11.4%) |
| | College | 748 (34.9%) | If drinking, frequency of drinking (n, %) | Once a month | 171 (70.1%) |
| | Postgraduate | 122 (5.7%) | | 2~4 times per month | 68 (27.9%) |
| Marital status (n, %) | Single | 2,121 (99.0%) | | 2~3 times per week | 5 (2.0%) |
| | Married | 21 (1.0%) | If drinking, the amount of drinking in a day (n, %) | 1 or 2 cups | 170 (69.7%) |
| Nationality (n, %) | Han | 2,048 (95.6%) | | 3 or 4 cups | 40 (16.4%) |
| | Hui | 35 (1.6%) | | 5 or 6 cups | 15 (6.1%) |
| | Manchu | 8 (0.4%) | | 7~9 cups | 7 (2.9%) |
| | Yi ethnic group | 8 (0.4%) | | ⩾10 cups | 12 (4.9%) |
| | Mongolian | 7 (0.3%) | | | |
| | Tujia | 7 (0.3%) | | | |
| | Others | 29 (1.3%) | | | |
| Grade (n, %) | Freshman year | 915 (42.7%) | Sleep duration (Median ± SD) | | 6.4 ± 1.0 |
| | Sophomore year | 702 (32.8%) | | <5 h | 116 (5.4%) |
| | Junior year | 285 (13.3%) | | 5~6 h | 449 (21.0%) |
| | Senior year | 80 (3.7%) | | 6~7 h | 1,055 (49.3%) |

*(continued on next page)*

**Table 1** (*continued*)

| Socio-demographic | Categories | Participants | Smoking, drinking and sleep condition | Categories | Participants |
|---|---|---|---|---|---|
| | University grade five | 38 (1.8%) | | 7~8 h | 460 (21.5%) |
| | Postgraduate grade | 122 (5.7%) | | 8~9 h | 46 (2.1%) |
| | | | | ≥9 h | 16 (0.7%) |
| | | | Nap duration (*n*, %) | 0~15 min | 580 (27.1%) |
| | | | | 16~30 min | 953 (44.5%) |
| | | | | 31~60 min | 514 (24.0%) |
| | | | | >60 min | 95 (4.4%) |

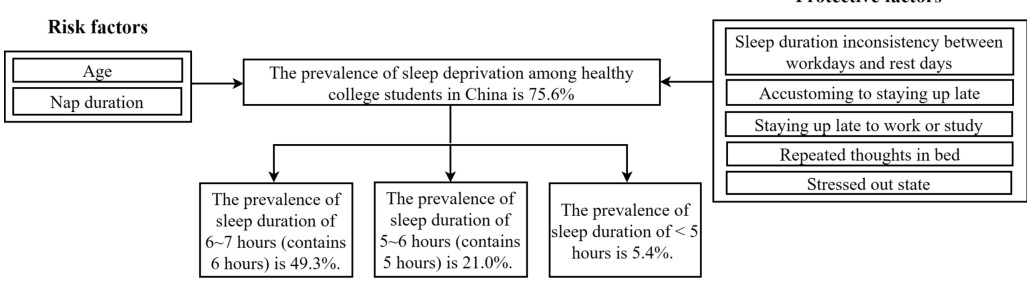

**Figure 1** **The results of prevalence and the factors of sleep deprivation among Chinese college students of the present study.**

In unadjusted models, the sleep deprivation was also associated with education, relationship issues, dormitory environment, mood problems, and desire to coordinate multiple life roles. Table 3 presents the results of crude odds ratio and adjusted odds ratio (A-OR) of binary logistic regressions on factors associated with sleep deprivation. The results are depicted in Fig. 1.

## DISCUSSION

To the best of our knowledge, this is the first study to report the prevalence of sleep deprivation among healthy college students in China. The prevalence of sleep deprivation estimated by the present study is the highest reported figure to date. More than three-quarters of the sample (75.6%) reported a sleep duration that did not meet the National Sleep Foundation's recommendations. Furthermore, this study also assessed the associated factors for sleep deprivation.

### Sleep deprivation is widespread among college students

The mean value of sleep duration of college students in this study was 6.4 ± 1.0 h. The National Sleep Foundation (NSF) recommends no less than 7 h per day of sleep for

**Table 2 Comparison of sleep-deprivation and non-sleep-deprivation groups among healthy college students in China.**

| | Sleep deprivation (N = 1,620) | Non-sleep deprivation (N = 522) | Z or χ² | P value |
|---|---|---|---|---|
| Age (median (IQR))[a] | 19.0 (17.0~20.0) | 19.0 (18.0~20.0) | −3.944 | 0.000*** |
| BMI (median (IQR))[a] | 21.0 (19.1~23.7) | 21.0 (19.0~23.4) | −0.935 | 0.350 |
| Education level (n(%))[b] | | | 4.998 | 0.082 |
| Junior college | 967 (59.7) | 305 (58.4) | | |
| College | 571 (35.2) | 177 (33.9) | | |
| Postgraduate | 82 (5.1) | 40 (7.7) | | |
| Grade (n (%))[b] | | | 9.442 | 0.088 |
| Freshman year | 696 (43.0) | 219 (42.0) | | |
| Sophomore year | 547 (33.8) | 155 (39.7) | | |
| Junior year | 212 (13.1) | 73 (14.0) | | |
| Senior year | 58 (3.6) | 22 (4.2) | | |
| University grade five | 25 (1.5) | 13 (2.5) | | |
| Postgraduate grade | 82 (5.1%) | 40 (7.7%) | | |
| Gender (n (%))[b] | | | 0.709 | 0.400 |
| Male | 441 (27.2) | 152 (29.1) | | |
| Female | 1179 (72.8) | 370 (70.9) | | |
| Marital status (n(%))[b] | | | 0.004 | 0.952 |
| Single | 1604 (99.0) | 517 (99.0) | | |
| Married | 16 (1.0) | 5 (1.0) | | |
| Nationality (n(%))[b] | | | 3.274 | 0.783 |
| Han | 1,547 (95.5) | 501 (96.0) | | |
| Hui | 24 (1.5) | 11 (2.1) | | |
| Manchu | 6 (0.4) | 2 (0.4) | | |
| Yi ethnic group | 6 (0.4) | 2 (0.4) | | |
| Mongolian | 6 (0.4) | 1 (0.2) | | |
| Tujia | 6 (0.4) | 1 (0.2) | | |
| Others | 25 (1.5) | 4 (0.8) | | |
| Smoking (n(%))[b] | | | 0.201 | 0.654 |
| No | 1,587 (98.0) | 513 (98.3) | | |
| Yes | 33 (2.0) | 9 (1.7) | | |
| If smoking, the number of cigarettes smoked per day (median (IQR))[a] | 5 (1.0~7.0) | 4 (3.0~14.0) | 0.402 | 0.695 |
| Drinking (n(%))[b] | | | 0.749 | 0.387 |
| No | 1,430 (88.3) | 468 (89.7) | | |
| Yes | 190 (11.7%) | 54 (10.3%) | | |
| If drinking, frequency of drinking ((n, %))[b] | | | 4.037 | 0.123 |
| Once a month | 133 (70.0%) | 38 (70.4%) | | |
| 2–4 times per month | 55 (28.9%) | 13 (24.1%) | | |
| 2–3 times per week | 2 (1.1%) | 3 (5.6%) | | |

**Table 2** (*continued*)

| | Sleep deprivation (*N* = 1,620) | Non-sleep deprivation (*N* = 522) | *Z* or $\chi^2$ | *P* value |
|---|---|---|---|---|
| If drinking, the amount of drunk in a day (*n* (%))[b] | | | 6.323 | 0.157 |
| 1 or 2 cup | 133 (70.0%) | 37 (68.5%) | | |
| 3 or 4 cup | 35 (18.4%) | 5 (9.3%) | | |
| 5 or 6 cup | 9 (4.7%) | 6 (11.1%) | | |
| 7–9 cup | 5 (2.6%) | 2 (3.7%) | | |
| ≥10 cup | 8 (4.2%) | 4 (7.4%) | | |
| Nap duration (*n*(%))[b] | | | 21.601 | 0.000*** |
| 0∼15 min | 459 (28.3) | 121 (23.2) | | |
| 16∼30 min | 728 (44.9) | 225 (43.1) | | |
| 31∼60 min | 378 (23.3) | 136 (26.1) | | |
| >60 min | 55 (3.4) | 40 (7.7) | | |
| Sleep duration on workdays and rest days (*n* (%))[b] | | | 6.398 | 0.040* |
| Same | 513 (31.7) | 172 (33.0) | | |
| Workdays > Rest days | 149 (9.2) | 66 (12.6) | | |
| Rest days > Workdays | 958 (59.1) | 284 (54.4) | | |

**Notes.**
[a]Mann–Whitney *U* test.
[b]Chi-square test.
IQR, interquartile range.
*P < 0.05.
***P < 0.001.

adult's ages 18–64 (*Hirshkowitz et al., 2015*). The mean sleep duration of college students in this study was below the recommended standard. In addition, the prevalence of sleep deprivation was 75.6% in this study, which indicates sleep deprivation is extremely common among college students. The prevalence of sleep deprivation among Chinese college students is close to the United States, and higher than Malaysia (58.1%) (*Naito, Yun Low & Yuen, 2021*), Pakistan (57%) (*Luqman et al., 2020*), Lebanon (52.7%) (*Assaad et al., 2014*), Japan (52%) (*Matsushita et al., 2014*), and Saudi Arabia (49.6%) (*AlFakhri et al., 2015*). The above data illustrates that the average sleep duration of college students decreased in both developed and developing countries. Two explanations were proposed for this phenomenon. First, due to the nation's rapid economic development, especially the establishment of logistical supply chains and internet-based technologies, which give people access to media and entertainment conveniently, people are sleeping shorter (*Fang et al., 2021*). Secondly, most Chinese college students are free from the supervision of their parents and start to live independently when they go to college. They prefer to sleep less when facing schoolwork and the abundant after-school life alone. Apparently, young people's lifestyles and behaviors have largely changed, and their sleep seems to be affected by rapid transformation and social changes (*Naito, Yun Low & Yuen, 2021*).

The prevalence of sleep deprivation among college students in China is significantly higher than Chinese population. *Fang et al. (2021)* reported the prevalence was 23.6%, 25.3%, 28.9%, 31.9%, and 33.1% in 2004, 2006, 2009, 2011, and 2015 respectively among the Chinese population. It suggests that sleep deprivation is more severe among college
**Table 3  Result of binary logistic regression on factors associated with sleep deprivation[a].**

| Variables | Crude OR (95% CI) | P | Adjusted OR (95% CI) | P |
|---|---|---|---|---|
| Age | 1.07 (1.03~1.12) | 0.001** | 1.05 (1.01~1.10) | 0.025* |
| Education | | | | |
|   Junior college | Reference | – | Reference | – |
|   College | 0.98 (0.80~1.22) | 0.873 | 0.92 (0.72~1.19) | 0.535 |
|   Postgraduate | 1.55 (1.04~2.31) | 0.032* | 0.98 (0.54~1.78) | 0.943 |
| Nap duration | | | | |
|   0–15 min | Reference | – | Reference | – |
|   16–30 min | 1.17 (0.92~1.51) | 0.212 | 1.14 (0.88~1.48) | 0.307 |
|   31–60 min | 1.37 (1.03~1.81) | 0.030* | 1.29 (0.97~1.74) | 0.085 |
|   >60 min | 2.76 (1.75~4.34) | 0.000*** | 2.35 (1.45~3.80) | 0.001** |
| Sleep duration on workdays and rest days | | | | |
|   Same | Reference | – | Reference | – |
|   Not the same | 0.55 (0.45~0.68) | 0.000*** | 0.61 (0.49~0.75) | 0.000*** |
|   Workdays >Rest days | 0.61 (0.42~0.87) | 0.006** | 0.63 (0.43~0.92) | 0.017* |
|   Rest days >Workdays | 0.54 (0.44~0.67) | 0.000*** | 0.61 (0.48~0.76) | 0.000*** |
| Accustomed to staying up late | | | | |
|   No | Reference | – | Reference | – |
|   Yes | 0.44 (0.35~0.55) | 0.000*** | 0.45 (0.36~0.57) | 0.000*** |
| Staying up late to work or study | | | | |
|   No | Reference | – | Reference | – |
|   Yes | 0.61 (0.49~0.76) | 0.000*** | 0.62 (0.49~0.78) | 0.000*** |
| Relationship issues | | | | |
|   No | Reference | – | Reference | – |
|   Yes | 0.64 (0.45~0.91) | 0.012* | 0.82 (0.57~1.20) | 0.310 |
| The dormitory environment affects sleep | | | | |
|   No | Reference | – | Reference | – |
|   Yes | 0.71 (0.56~0.91) | 0.006** | 0.89 (0.68~1.15) | 0.358 |
| Stressed out state | | | | |
|   No | Reference | – | Reference | – |
|   Yes | 0.63 (0.49~0.80) | 0.000*** | 0.75 (0.58~0.98) | 0.036* |
| Repeated thoughts in bed | | | | |
|   No | Reference | – | Reference | – |
|   Yes | 0.71 (0.57~0.89) | 0.002** | 0.79 (0.62~0.99) | 0.044* |
| Mood problems | | | | |
|   No | Reference | – | Reference | – |
|   Yes | 0.64 (0.47~0.88) | 0.006** | 0.89 (0.62~1.28) | 0.534 |

**Table 3** (*continued*)

| Variables | Crude OR (95% CI) | *P* | Adjusted OR (95% CI) | *P* |
|---|---|---|---|---|
| Desire to coordinate multiple life roles | | | | |
| No | Reference | – | Reference | – |
| Yes | 0.76 (0.59~0.99) | 0.044* | 0.88 (0.67~1.16) | 0.357 |

**Notes.**

Reference is the reference variable.

OR, odds ratio; unadjusted OR, crude OR; 95% CI, 95% confidence interval.

[a]Binary logistic regression after adjusted OR, Hosmer-Lemeshow $\chi 2 = 5.491$, $P = 0.704$; Nagelkerke $R^2 = 0.097$.

*$P < 0.05$.

**$P < 0.01$.

***$P < 0.001$.

students. In this regard, we have analyzed several possible reasons. Firstly, freshmen and sophomores face new challenges, such as being solely responsible for themselves, new schedules, unfamiliar environments, social obligations, roommate disagreements, dietary problems, financial burdens, and academic stress (*Luqman et al., 2020*). The senior students confront with graduation, career choice and master's entrance examinations. For the postgraduate group, it is common to stay up late for scientific work or paper writing based on our observations. In addition, graduate students also have to face the pressure of income, marriage and childbirth due to their age. These factors above force students to reduce the duration or change the habits of sleep, resulting in sleep deprivation (*Trockel, Barnes & Egget, 2000*). Secondly, the majority of students living in dormitories had a later bedtime and early wake-up time, longer sleep latency, and shorter total sleep duration (*Lima, Medeiros & Araujo, 2002*). Thirdly, based on the researcher's observations and previous studies (*Christensen et al., 2016*; *Hale & Guan, 2015*; *Whiting et al., 2021*; *Mireku et al., 2019*; *Albqoor & Shaheen, 2021*), it was shown that some college students stay up even all night for entertainment and socializing, such as playing games, watching movies, TV shows or entertainment programs, shopping online, chatting on cell phones, *etc.* These facts may explain current the higher incidence of sleep deprivation among college students in China.

The prevalence of sleep deprivation among college students is significantly higher than adolescents in China. Previous studies showed that the prevalence of sleep deprivation among Chinese adolescents was 51% (*Chen et al., 2014*). Compared to adolescent students, college students are less likely to have parents monitoring their sleep duration (*Noor et al., 2022*). Therefore they have more time (at the expense of sleep) to do what they want to do. The prevalence of sleep deprivation among college students is similar to some high-pressure work groups. For instance, the working population was 71.3% (*Visvalingam et al., 2020*), and the military was 72% (*Luxton et al., 2011*). Among the groups with high prevalence of sleep deprivation, some unavoidable factors may contribute to the occurrence of sleep deprivation, such as commuters staying up late and getting up early to work (*Visvalingam et al., 2020*), and military drills for soldiers regardless of the time of day (*Luxton et al., 2011*). For most college students, getting up early for class and sleeping late may be a reason for the higher prevalence of sleep deprivation.

The prevalence of sleep deprivation in this study was 55.0% in females and 20.6% in males. Previous studies have shown similar results: Becker report showed that female college students had a higher prevalence of sleep deprivation than male (*Becker et al., 2018*).
It should be reminded that more female participants than male in the sample may have contributed to the higher prevalence of female sleep deprivation. However there was no statistical difference in the prevalence of the two groups. In addition, there were more women who slept normally in this study.

## Factors associated with sleep deprivation

Healthy sleep is extremely important for college students. Chronic sleep deprivation may have an impact on cognition (memory, attention, alertness, *etc.*) (*Alhola & Polo-Kantola, 2007*), mental state (*Liu et al., 2013*), physical activity (*DePorter, Coborn & Teske, 2017*), and more. The ability to prevent adverse effects depends on identifying the relevant factors and implementing the necessary interventions to minimize the growth rate of sleep deprivation. Our study also identified the risk and protective factors for sleep deprivation.

### Sleep deprivation may have been caused by an increase in age and nap duration

Our results indicated the risk of sleep deprivation increases with age, which was consistent with previous studies. *Fox et al. (2018)* have demonstrated that sleep deprivation was more likely to occur with age. The study by *Ohayon et al. (2004)* had shown that several sleep-related factors decreased with age, including the total sleep time, sleep efficiency, slow-wave sleep, and rapid eye movement sleep. Several possible explanations might be related to the phenomenon. First, a reduction in sleep duration with age could be attributed to changes in sleep structure and a reduction in acute sleep needs (*Skeldon, Derks & Dijk, 2016*; *Lee & Sibley, 2019*). Also, the study by *Grandner (2022)* had shown that sleep becomes more erratic, as age increasing. The majority of the sample in this study was from medical universities. Seniors need to deal with heavier coursework or clinical practice. This leads to the possibility that they may sacrifice some sleep time to study or work. In addition, some college students use their bedtime to relax and release the tension of the day at night (*Acharya, Acharya & Waghrey, 2013*). The seniors have psychological stress in areas such as job hunting. They tend to sleep less to meet higher demands in academics, facing fierce social competition.

The current study found that napping for more than 60 min per day increased the risk of sleep deprivation. This finding was consistent with that of *Dhand & Sohal (2006)* who suggested that naps of less than 30 min confer several benefits, whereas longer naps might reduce productivity and sleep inertia and affect physical health. The study by *Dhand & Sohal (2006)*, *Mead et al. (2022)* and *Hays, Blazer & Foley (1996)* showed that long naps might reduce subsequent night sleep. Although daytime napping was thought to delay sleep and contribute to shorter nighttime sleep, it had also been found to be a mechanism to compensate for sleep deprivation during weekdays and weekends (*Gradisar et al., 2008*; *Malone et al., 2016*). However, some studies (*Monk et al., 2001*; *Pilcher, Michalowski & Carrigan, 2001*; *Campbell, Murphy & Stauble, 2005*) did not find long naptime affecting nighttime sleep, which differs from above findings. Therefore, the increased risk of sleep deprivation due to long naps needs to be further explored.

### Sleep duration inconsistency between workdays and rest days, accustoming to staying up late, staying up late to work or study, repeated thoughts in bed, and stressed out state are protective factors of sleep deprivation

Compared with sleeping consistency, sleeping inconsistency between work and rest days had 40% lower odds of sleep deprivation. It suggests sleeping inconsistency between work and rest days might be a protective factor for sleep deprivation. For people who are inconsistent, more sleep is an important way to restore energy and strength. Getting more sleep on rest days can ease the strain of the workday. As *Lund et al. (2010)* reported that the rest day's wake-up time could be delayed due to lack of pressure to go to class. *Zhang et al. (2020)* found that longer rest days catch-up sleep appeared to help adults recover from moderate sleep loss on workdays. For students who sleep more on workdays, workday routine may allow them to get enough sleep. Therefore they may use more time to relax on their rest days. Less sleep is likely to be no less than 7 h, both on weekdays and on rest days.

The results of this study showed that those accustomed to staying up late and staying up late to work or study were less likely to have sleep deprivation. This finding was unexpected and not the same as previous research (*Ahrberg et al., 2012*; *Orzech, 2013*; *Gariépy et al., 2019*). One possible reason is that the students who stayed up late often had a correspondingly delayed wake-up time, and their overall sleep time did not decrease. This sleep pattern is similar to the healthcare workers who worked at home during the COVID-19 pandemic. Some students even sleep longer to compensate for the effects of sleeping late.

In general, stress and repeated thoughts in bed may affect sleep. Such as *Almojali et al. (2017)* reported that students who were not suffering from stress were less likely to have sleep problems. One unanticipated finding detected by this study was stress and repeated thoughts in bed reducing the risk of sleep deprivation. Two possible reasons may explain this result. The subjects of this study were healthy college students without psychiatric and psychological problems. Such individuals are able to relieve stress through exercise, recreation, and socialization. They are able to tolerate stress in normal limits without developing sleep disorder. In addition, school closure management due to COVID-19 pandemic may explain this phenomenon. A study from a US college student population showed the sleep duration was significantly longer, bedtime and wake time were significantly later during the COVID-19 pandemic (*Benham, 2021*), which indicated that COVID-19's stress on sleep duration may not be entirely negative. A study from India showed relative to the before lockdown condition, undergraduate and postgraduate university students were more extensive feelings of sleepiness, with significantly increased daytime nap duration, and depressive symptomatology, during the phase of the pandemic (*Majumdar, Biswas & Sahu, 2020*). Similar to that, our investigation was conducted during the COVID-19 pandemic. During the period, the school was in closed management and students may feel stressed. On the other hand, due to online learning, students were relatively free to spend their time. They had not to wake up on time due to strict class time, which gave them the ability to sleep longer. As *Benham (2021)* reported students stay up late and also get up late.

Therefore the sleep duration is likely to be adequate. In addition, there may be some other potential factors that influenced the results. We are going to conduct semi-quantitative personal interviews with the aim of finding evidence that would explain this result.

## Measures should be undertaken to promote a healthy sleep

It is necessary to perform methods maintaining enough sleep due to the current high incidence of sleep deprivation among college students. On an individual basis, the daytime hours should be rationalized, and the nap duration should be controlled; get enough sleep on rest days to replace missing hours of sleep on workdays; posters about "the importance of adequate sleep" or "going to bed" in the dorm could be put up; a time for lights out is set negotiating with your roommates; the electronics are not bring to bed. On a college basis, administrators could try measures to make sure students have a good sleep-work balance. The special "lunchroom" is provided to encourage students napping. Sleep education activities could be conducted, such as carrying out sleep health lectures or blackboard competitions. Credit courses with sleep self-assessment are offered. The emails with the content of sleep relaxation techniques, a self-administered sleeps restriction protocol, *etc.* could be sent to students regular weekly by college. The special sleep teams could be set up to counsel students with sleep problems and make regular return visits by college (*Hershner & Chervin, 2014*).

Finally, it is important to indicate that our study has some limitations. This is a cross-sectional design. It was difficult to determine true exposure-outcome relationships. Therefore, the results are hypothesis generating and further large scale longitudinal studies are needed to validate our findings. Furthermore, the question assessing sleep duration and health status based on self-reported and responses are subjective. Self-reported sleep duration may be sometimes imprecise due to night-to-night differences in sleep and difficulty in distinguishing accurate sleep duration from time in bed. Although the students determined the health status according to the physical examination and a psychological assessment conducted annually. Subjective measurement methods are prone to measurement and recall bias, which can affect the validity and accuracy of reported sleep durations and health status. Therefore, in future research, the preferred method to accurately assess sleep duration is an objective measurement such as sleep actigraphy or polysomnography. Furthermore, the lack of a universal definition for defining sleep deprivation means that different studies use different sleep duration to define the outcome, making comparisons between studies difficult.

## CONCLUSIONS

This study revealed a high prevalence of sleep deprivation among healthy college students in China. Also, the study examined socio-demographics and factors that may have an impact on sleep comprehensively. Sleep deprivation may have been caused by an increase in age and nap duration. Sleep duration inconsistency between workdays and rest days, accustoming to staying up late, staying up late to work or study, repeated thoughts in bed, and stress may be protective factors of sleep deprivation which needs to be confirmed by longitudinal studies or personal interviews. Considering the current situation of widespread sleep

deprivation among college students, it is necessary to implement multi-party interventions in cooperation with society, schools, and college students themselves. Such as rationalizing daytime hours, controlling nap duration, getting enough sleep on rest days to replace missing hours of sleep on workdays. Promoting college students and public awareness of the importance of adequate sleep and the consequences of sleep deprivation would be the basic way to solve the high prevalence of sleep deprivation.

## ACKNOWLEDGEMENTS

We would like to thank the teacher Nai Bao Hu from Binzhou Medical University, who provided the application of statistical to analyze data.

### Funding

This work was supported by the National Nature Science Foundation (81703983). The funders had no role in study design, data collection and analysis, decision to publish, or preparation of the manuscript.

### Grant Disclosures

The following grant information was disclosed by the authors:
National Nature Science Foundation: 81703983.

### Competing Interests

The authors declare there are no competing interests.

### Author Contributions

- Congcong Guo analyzed the data, authored or reviewed drafts of the article, and approved the final draft.
- Songzhe Piao analyzed the data, authored or reviewed drafts of the article, and approved the final draft.
- Chenyu Wang analyzed the data, prepared figures and/or tables, and approved the final draft.
- Lili Yu performed the experiments, prepared figures and/or tables, and approved the final draft.
- Kejun Wang analyzed the data, prepared figures and/or tables, and approved the final draft.
- Qian Qu performed the experiments, prepared figures and/or tables, and approved the final draft.
- Cuiting Zhang analyzed the data, prepared figures and/or tables, and approved the final draft.
- Xiaofei Yu conceived and designed the experiments, authored or reviewed drafts of the article, and approved the final draft.

## Human Ethics

The following information was supplied relating to ethical approvals (*i.e.*, approving body and any reference numbers):

The study was conducted according to the guidelines of the Declaration of Helsinki and approved by the Ethics Committee of Binzhou Medical University (2021-262).

## Data Availability

The raw measurements are available in the Supplementary File.

## Supplemental Information

Supplemental information for this article can be found online at http://dx.doi.org/10.7717/peerj.16009#supplemental-information.

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
