# Peer review of "The prevalence and associated factors of sleep deprivation among healthy college students in China: a cross-sectional survey"

_PeerJ, doi:10.7717/peerj.16009_

## Round 0.1 · original submission · Major Revisions

Please address the concerns raised by the reviewers.

Reviewer 1 ·

Basic reporting

no comment

Experimental design

no comment

Validity of the findings

The authors have done a great job framing this interesting study. However, there are some major concerns.
- The healthy cohort should be compared with the cohort with mental illness or physical illness to enhance the strength of the study.
- Although the results are statistical significant, it does not translate clinically. The findings are not confirmatory. Needs further validation.
- Other factors need to be taken into consideration for sleep deprivation and need to be adjusted in analysis like caffeine drinks, recreational drugs, and stimulants.
- The novelty of this study needs to be highlighted. How it is different from the existing studies?
- Depicting the study findings in one image as central illustration would be beneficial.
- Rest comments as mentioned in the manuscript.

Annotated reviews are not available for download in order to protect the identity of reviewers who chose to remain anonymous.

Reviewer 2 ·

Basic reporting

no comment

Experimental design

no comment

Validity of the findings

no comment

Additional comments

no comment

·

Basic reporting

Minor grammatical errors and punctuation errors.

Experimental design

The study design is based on questionnaires which are sometimes difficult to validate and replicate. However, it could aid in further investigation and future studies.
Overall well done.

Validity of the findings

no comment

Additional comments

Well-written article. Few comments.

1. The study has higher female participants, so naturally, the prevalence will be higher in the results, which can skew the findings.

2. [277 Sleep duration inconsistency between workdays and rest days, accustoming to staying up
278 late, staying up late to work or study, repeated thoughts in bed, and stressed out state are
279 protective factors of sleep deprivation.] The statements contradict the known factors of sleep deprivation.

---

## Round 0.2 · accepted · Accept

The original Academic Editor is not available so I am making the decision in my capacity as Section Editor,

Thank you for all revisions made to this manuscript

Reviewer 1 ·

Basic reporting

no comment

Experimental design

no comment

Validity of the findings

no comment

Additional comments

no comment